# Embedding an Evidence-Based Model for Suicide Prevention in the National Health Service: A Service Improvement Initiative

**DOI:** 10.3390/ijerph17144920

**Published:** 2020-07-08

**Authors:** Sophie Brown, Zaffer Iqbal, Frances Burbidge, Aamer Sajjad, Mike Reeve, Victoria Ayres, Richard Melling, David Jobes

**Affiliations:** 1Department of Psychology, Faculty of Health Sciences, University of Hull, Hull HU6 7RX, UK; S.Brown@2017.hull.ac.uk (S.B.); f.burbidge@hull.ac.uk (F.B.); 2NAViGO Health and Social Care Community Interest Company, Grimsby DN32 0QE, UK; aamer.sajjad1@nhs.net (A.S.); michael.reeve@nhs.net (M.R.); vicky.ayres@nhs.net (V.A.); richard.melling@nhs.net (R.M.); 3Department of Psychology, School of Arts and Sciences, Clinical Psychology Faculty, The Catholic University of America, Washington, DC 20064, USA; jobes@cua.edu

**Keywords:** suicide prevention, suicidality, CAMS, service improvement, service model

## Abstract

Despite the improved understanding of the determinants of suicide over recent decades, the mean suicide rate within the United Kingdom (UK) has remained at 10 per 100,000 per annum, with about 28% accessing mental health services in the 12 months prior to death. In this paper, we outlined a novel systems-level approach to tackling this problem through objectively differentiating the level of severity for each suicide risk presentation and providing fast-track pathways to care for all, including life-threatening cases. An additional operational challenge addressed within the proposed model was the saturation of local crisis mental health services with approximately 150 suicidality referrals per month, including non-mental health cases. This paper discussed a service improvement initiative undertaken within a National Health Service (NHS) secondary care mental health provider’s open-access 24/7 crisis and home treatment service. An organisation-wide bespoke “suicide risk triage” system utilising the Collaborative Assessment and Management of Suicidality (CAMS) was implemented across all services. The preliminary impacts on suicidality, suicide rates and service user outcomes were described.

## 1. Introduction

On average, someone dies by suicide every 40 s somewhere in the world [1]. Many more people die by suicide each year than in road traffic accidents, yet the funding for suicide prevention is significantly lower in comparison to road accident prevention [2], with the economic cost of suicide estimated at £1.7 million per individual [3]. Recent findings indicate that over 100 people are affected by every single suicide [4], with an increased likelihood of suicidal ideation and poor psychiatric outcomes for those closest to the individual [5]. Further, one in five adults in England report experiencing suicidal thoughts at some point in their lifetime [6]; broadening the focus of clinical treatments to include this larger population may have implications for reducing morbidity [7]. Historically, and in spite of the focus on suicide prevention in national policy, suicide rates have remained high, suggesting that certain components of the UK’s national strategy require elucidation.

An important issue for suicide prevention is to identify those at risk of suicide and direct treatment efforts accordingly to prevent these individuals from taking their own lives [8]. Nationally, mental health organisations have improved their ability to quantify the nature and extent of all types of suicidality presentation [9]. National Health Service (NHS) mental health trusts provide a range of government-funded inpatient and community care for a locality [10]. The specialist provision includes acute psychiatric care and specialist services for child and adolescent, older peoples and learning disabilities, as well as early intervention in psychosis teams, forensic and drug and alcohol services, through to primary care programmes that work alongside family care doctors in healthcare centres. NHS mental health services are free and often require a doctor’s referral, although an increasing number of services can be accessed via self-referral [10]. In 2000, Crisis Resolution and Home Treatment Teams (CRHTTs) were introduced in England, including 24-h availability and intensive support for individuals experiencing a mental health crisis [11]. Despite the implementation of CRHTTs offering 24/7 support, suicide rates have shown little variation since 2008. This is, in part, a reflection of the difficulties associated with the historical focus on accurate suicide risk prediction. Risk assessment scales would be commonly used in clinical practice to quantify the risk of suicide, with 85% of NHS mental health trusts using checklist-style approaches [12] and approximately two-thirds using locally devised adaptations that lacked formal psychometric validation [13]. Recent opinion has confirmed the limited clinical utility for predicting suicide and self-harm using risk assessment scales [14,15,16], and that the use of such scales may result in unnecessarily restrictive treatment options for those categorised as “high-risk” [17]. Evidence reviewing the predictive value of widely used risk scales in the UK has highlighted the low specificity of such scales for suicide and self-harm, which may result in individuals remaining within mental health services for longer than necessary [14,18]. In such cases where false positives for suicide risk are identified, targeted treatment to assist suicidality may be superseded by more restrictive care planning, such as compulsory detainment and hospitalisation [18].

Given that suicidality does not exclusively exist in the context of pre-existing mental health issues, an additional challenge for suicide prevention is appropriate provision for those with non-mental health-related needs [19]. Individuals with suicidality present with a variety of needs that are not exclusively mental-health-based, including societal, community, relationship and individual risk factors [20]. Assuming that suicidality is the result of a mental health diagnosis may place an unnecessary burden on mental health professionals to prevent suicide, as well as increased blame if an individual who does not seek help completes suicide [21]. Previous research estimates that, for those individuals who do have contact with healthcare services, only between 3% and 22% of individuals had reported suicidal intent at their final appointment with a healthcare professional before their suicide [22,23,24], suggesting suicide risk identification is more complex than a simple dyadic relationship between suicide expression and psychiatric disorder. It is therefore unsurprising that national suicide rates have remained high, given the limited utility of suicide risk prediction methodologies that have remained commonplace across mental health providers.

A consequence of these prediction methodologies, whilst suicide rates remain consistently high, will be their impact on clinicians’ confidence when assessing suicide risk. Dealing with patients who self-harm and/or are suicidal is perhaps one of the most difficult challenges faced by clinicians [25]. One study estimated that 88% of mental health professionals have at least some level of fear relating to a patient dying by suicide, as well as discomfort around working with suicidal patients [26]. The limited training that mental health professionals receive relating to the assessment and management of suicidality may contribute to the burden felt by clinicians working in healthcare settings [27]; in some cases, a form of helplessness as suicide rates remain unaffected and predictive data have little impact on reversing this rate. The checklist-style structure of risk assessment within many NHS mental health services forms an “aide memoire” of items characteristic of many suicide risk prediction tools. Clinicians’ anxieties may increase the reliance on undertaking an assessment based upon a checklist of phenomenological or epidemiologically valid items that provide few opportunities to account for individual differences that may provide a more accurate and richer suicide risk assessment [14]. Thus, the use of such tools may provide a false reassurance of safe and effective working, appease the clinician’s burden and sense of dyscontrol, whilst giving the impression of effective working and so mediating corporate risk. Furthermore, losing a patient by suicide can impact on professional practices, including issues around objective clinical decision-making, increased vigilance when dealing with future suicidal patients and avoidance of treating suicidal patients [28,29], almost a form of learned helplessness where reliance on the same systems for assessment and treatment continues. However, evidence highlights that training focusing specifically on the management of the suicidal drivers, or factors mediating the cognitions, emotions and behaviours augmenting suicidal risk, resulting in suicidal behaviours, can have a positive effect on clinicians’ confidence, clinical skills and implementation of evidence-based practices [30,31]. One such approach is the Collaborative Assessment and Management of Suicidality (CAMS), a promising Randomised Controlled Trial (RCT) evidence-based intervention that aims to target and treat the factors that maintain suicide risk [32,33]. The core empirical component of CAMS, the Suicide Status Form (SSF), is a multidimensional tool used for comprehensive assessment, treatment planning and tracking of suicide risk [32]. Patient-defined direct and indirect suicidal drivers are identified within the SSF. Direct drivers are psychologically-based components that are idiosyncratic, whereas indirect drivers are stressors, such as housing issues, relationship difficulties or unemployment [34]. Indirect drivers may not necessarily result in suicidality but can increase vulnerability to the direct drivers of suicide. The SSF core assessment, a quantitative section completed by the patient, includes five psychological constructs (psychological pain, stress, agitation, hopelessness and self-hate) and has established reliability and validity [32]. This is complemented by a qualitative assessment of the psychological drivers and empirically-derived risk factors implicated in self-harm and suicidality. The resulting treatment plan, using therapeutic inputs pertinent to the defined risks from the SSF core assessment, is reviewed in ongoing tracking sessions until the risk of suicide has abated. The CAMS intervention ends after three consecutive sessions of successfully managing suicidal thoughts, feelings and behaviours, thus lasts a minimum of four sessions, with an average number of 12 sessions [35]. RCT results highlight the positive impacts of CAMS on reducing suicidal ideation, overall symptom distress and instilling increased hope for individuals presenting with suicidality across a variety of settings [35,36,37]. In addition, evidence suggests that CAMS is as effective as Dialectical Behaviour Therapy (DBT) for reducing suicide attempts and self-harm behaviours [38]. There is also evidence that CAMS training can significantly decrease clinician’s anxiety about working with suicidal risk and increase confidence, with results sustained at 3-month follow-up [39]. The CAMS approach has yet to be evaluated in the UK; the impact of the CAMS intervention for service users presenting with life-threatening behaviours within the current research will be evaluated in a follow-up paper.

### The Current Research

Clinicians’ confidence when working with suicide risk is a pertinent issue for suicide prevention [40], with a tendency for clinicians to focus on predicting the probability of suicide despite little evidence supporting the utility of this approach [14,15,16]. Given the issues around risk prediction and clinician confidence, the NHS mental health provider where the research was being undertaken implemented a service-wide, systems-level approach to suicidal risk (referred to hereon as “suicide risk triage”). The “suicide risk triage” model was a hierarchically supervised, individual-specific real-time suicide risk assessment and care planning process that seeks to assess suicide risk and intervene accordingly, including rapid access to a range of evidence-based treatments for individuals presenting with life-threatening behaviours. “Suicide risk triage” preceded the CAMS intervention, which was the central treatment for acute suicidality within the model. The CAMS intervention was utilised for all high-risk suicidality presentations, which represented a small proportion of all cases (less than 2%) as the majority of suicide risk presentations within the current research required low-level support or signposting to deal with social care and relationship issues (indirect drivers). Although the CAMS intervention can be used for all levels of suicidality [7], resource implications would overwhelm mental health services if attempting to utilise CAMS for all suicidality presentations (over 4000 presentations since the start of the research project). For most cases where a “suicide risk triage” was completed for a service user, this resulted in emotional and practical support provision, given that they were experiencing rapid emotional dysregulation due to a life event, and their suicidality was resolved within a matter of days. Utilising the CAMS intervention for this cohort would arguably be a less efficient use of resources. As such, the results described related to the impact of “suicide risk triage” rather than the CAMS intervention itself, which would be disseminated at the project’s completion. To summarise, then combined with a supervision hierarchy, the “suicide risk triage” aimed to address the concerns of all clinical staff, in order to minimise confounders of objective, person-specific clinical risk decision-making and thus increasing the probability of effective management of suicide risk presentations.

This paper presented an outline and evaluation of a systems-level “suicide risk triage” model and preliminary impacts on suicidality, suicide rates and service user outcomes. The main objectives were to reduce the number of suicides in the locality, improve the service user experience for individuals presenting as at high risk of suicide (and offer rapid access treatment options) and improve clinicians’ confidence when dealing with such cases.

## 2. Materials and Methods

### 2.1. Research Site and Team

This study was carried out within an NHS secondary care mental health service provider, based in the North of England and serving a population of approximately 158,000 spread across a semi-rural area encapsulating three small towns and with service provisions equivalent to other NHS mental health trusts. Several known risk factors for suicide exist within the locality with the most prevalent being substance misuse, relationship problems, social isolation, long-term unemployment and physical health difficulties [41]. The local provision includes acute inpatient services, crisis and home treatment support, older adult inpatient and memory services, community mental health services as well as a range of specialist teams supporting adults with mental health difficulties, such as psychosis, eating disorders and personality disorders. In addition to these services, the provider services met the National Institute of Health and Care Excellence (NICE) guidelines and quality standards for all Axis 1 and 2 disorders that it is commissioned to provide.

The research was undertaken at the local psychiatric inpatient hospital site where crisis and home treatment services are provided with approximately 150 referrals per month. The open-access crisis and home treatment service is available through self-presentation or via phone 24 h a day, 7 days a week in line with national guidance [11] and are gatekeepers to acute inpatient beds. External referrals to the crisis team are through various pathways, including primary care, general hospital and other teams within secondary care mental health services. The crisis team comprised of 23 qualified clinicians primarily from social work and nursing backgrounds with an average of 8 years’ experience in mental health services. An additional sub-team of 10 clinicians also worked at the local general hospital site to provide mental health crisis and liaison support for individuals presenting to the Accident and Emergency department with suicidality. All presentations to these services received a comprehensive assessment of health and social care needs, including psychiatric history, current mental health symptomology (if any), social functioning, risk and support networks. Clinicians conducted a detailed assessment and provided recommendations, which included one or more of the following based on the individual’s needs: an inpatient admission, medication review, intensive home treatment input, referral to community services, social prescribing and/or signposting to third sector organisations as appropriate. Several links with numerous external providers, including substance misuse services, housing agencies and domestic violence support, were well established to ensure a smooth transition to these services should they be required.

### 2.2. “Suicide Risk Triage”

A systems-level “suicide risk triage” model was implemented to objectively differentiate the most severe suicide risk presentations and provide appropriate care for all cases irrespective of the severity of intent. “Suicide risk triage” preceded and further facilitated all clinical risk decisions when a service user expressed suicidality, suicidal intent or where a clinician had concerns that such risk might be evident. The process involved the clinician making a decision about the level of suicidal risk when a service user disclosed suicidal ideation, intent or behaviours and identifying whether there is a need for social and practical support, identification of relapse in psychiatric cohorts, or additional risks due to self-harm or life-threatening behaviour. Clinicians were given detailed guidance around the pertinent variables to consider when assessing the level of suicide risk, including history of suicide/self-harm, medical and surgical treatment needs (if any), help-seeking behaviours, the pathway to services and current presentation. This information was collated and inputted on the service user’s electronic record system, which consisted of further questions regarding the primary driver for the service user’s suicidality, the identified treatment option, the rationale for the treatment option and whether additional supervision was necessary for the assessing clinician to make a decision regarding the level of suicide risk/care planning.

Recognition of the issues around clinicians’ confidence when assessing suicide risk resulted in the development of a real-time hierarchical supervision component. This was derived to address the themes relating to clinicians’ anxieties when working with suicidal patients, identified through training sessions, which highlighted the value of shared responsibility with senior supervising colleagues when considering more challenging suicide risk assessments. Within the “suicide risk triage” model, a supervision hierarchy was set up across the organisation to support clinicians if they were unsure about the level of suicidal risk a service user presented with, the treatment plan they would develop for them, or if they felt that the risk was potentially life-threatening and therefore needed escalation for a CAMS assessment (Washington, DC, USA) and intervention. The supervision structure included additional training for nominated clinicians within each team who were available to support/advise their colleagues when making difficult decisions around assessment and management of suicide risk. This support could be extended further up the hierarchy to CAMS-trained clinicians and senior staff with extensive experience of managing clinical risk.

For the majority of cases, clinicians agreed that they would feel able to formulate the clinical variables pertaining to risk, make a triage decision and care plan accordingly. However, if a clinician was unclear about the severity of the suicidal risk, the supervision system would be accessed. This involved recording their opinion within the electronic record system followed by an immediate face-to-face or telephone discussion with the next level of the hierarchy, so as to further elucidate the clinical picture pertaining to the individual service user and thus establish more accurately the extent of suicidal risk. The hierarchy would continue to be accessed until a consensus decision was reached.

### 2.3. Training

Three phases of training were delivered across the organisation: “suicide risk triage” training, CAMS training, and CAMS concordance.

Phase I: All qualified staff were required to attend a mandatory 1-day training course entitled ‘risk triage training’ in groups of approximately 12, resulting in 320 trained mental health clinicians. Besides this training providing an overview of how the “suicide risk triage” model was to be implemented within services, it also allowed for the collation of variables (through breakout focus groups of 3–4 individuals) that clinicians felt impacted on their confidence during the suicide risk assessment. The focus groups were asked to discuss the anxieties and concerns they had when undertaking a suicide risk assessment. Feedback from each group was collated into themes that were addressed within the training provided to those providing supervision at higher levels of the hierarchical suicide triage process. Finally, the training also ensured all clinicians met a baseline level of ability and knowledge and was delivered to all new and newly qualified clinicians.

During these training sessions, clinicians expressed a range of concerns relating to suicide risk decision-making, including the impact of the organisation’s NHS-specific investigation process (utilising root-cause analysis methodology) following a completed suicide, obtaining accurate information in order to make objective decisions around suicide risk and support from other colleagues/managers when making these decisions. The supervision hierarchy was implemented to mitigate the effects of these concerns by ensuring that any serious incidents relating to suicidal behaviours within the organisation would become the collective responsibility of all levels of the hierarchy, from frontline clinicians to senior and executive managers. Pre- and post-training evaluations addressing clinical practices when working with suicidal risk, level of comfort treating suicidal patients and the training delivery were completed by clinicians prior to and following triage training [39]. Analysis of these evaluations would be used to assess the extent of the project’s impact on clinicians’ confidence. Anecdotal feedback from the training highlighted the positive impact of a clear, structured approach to clinical risk decision-making to help clarify the most appropriate pathways to care for suicide risk presentations and the benefit of having support available for decision-making around challenging risk cases. Consequently, a psychometric tool with items utilised from the triage training was being developed as a measure of clinicians’ confidence.

Phase II: All members of the crisis/home treatment and mental health liaison teams, plus operational leads, were trained in CAMS, totalling 21 trained clinicians. The CAMS intervention was part of the “suicide risk triage” model and was utilised for service users presenting with life-threatening behaviours, representing less than 2% of all suicidality presentations within this research project. CAMS training consisted of the aforementioned “risk triage training” workshop, followed by a 3-h CAMS online video providing an overview of the CAMS assessment and a clinical demonstration of using CAMS with a patient. The video demonstrated the key techniques and components of the CAMS framework with an emphasis on remaining suicide-focused and working collaboratively with the service user.

Phase III: CAMS-trained clinicians were required to undertake a CAMS assessment with a service user observed by one of the project leads. Adherence to the model was assessed using the CAMS Rating Scale (CRS.3) [33]. Operational leads were responsible for ensuring continued adherence to the CAMS framework through supervision of all CAMS cases throughout the duration of the research project. A purposive sample of clinicians was interviewed regarding their acceptability of the CAMS model.

### 2.4. Supervision Hierarchy

The organisation’s unique risk decision-making process supporting the CAMS intervention included a supervision hierarchy. A 4-level hierarchical structure was set up across the organisation to support clinicians if they were unsure about the level of suicidal risk a service user presented with, or if they felt that the risk was potentially life-threatening and therefore needed escalation for a CAMS assessment. Thus, joint “ownership” of risk decisions was available whenever a clinician believed this was required and extended to senior and executive clinical and managerial staff. Four levels comprised the supervision hierarchy for the triage process:Level 1: All individual cliniciansLevel 2: Nominated departmental (clinical team) championsLevel 3: CAMS-trained clinicians and CAMS project leads (two senior clinicians)Level 4: Medical, clinical (principal investigator) and operational leads were all CAMS trained.

Departmental champions received additional training to help differentiate between life-threatening behaviour and self-harm to provide supervision within their teams.

### 2.5. Electronic Recording and Outcome Measures

“Suicide risk triage” decisions were inputted on the service user’s record as part of an electronic form where the clinician documented if the service user’s suicidal ideation/intent would suggest life-threatening behaviour, self-harm/Non-Suicidal Self Injury (NSSI)/relapse, required primary care service input (rather than secondary care) or whether the current support/management process was appropriate, as well as providing a rationale for the decision. Clinicians also used the electronic form to document if they were unclear about the service user’s level of suicide risk and recorded discussions with colleagues within the supervision hierarchy.

Data capture was set up via an electronic recording system, which collated real-time data about triages, including numbers per team and demographic information, to help analyse trends in triage data. Additionally, demographic and clinical data about CAMS assessments were updated every 24 h and used to monitor data collection for the experimental group as well as to assist with missing data checks.

Data on local suicide rates were a key part of the research project to assess whether any changes were observed for both general population and mental health patient suicides, although longitudinal data at the end of the 3-year project term would be required to assess whether reductions were part of a maintained trend. The key outcome measures for service user presentations via the “suicide risk triage” model were on service utilisation, i.e., future crisis/acute service provision and continued engagement with services. Qualitative interviews were conducted specifically with those service users who had undertaken the CAMS intervention and with clinicians at various levels on the supervision hierarchy and thus a breadth of involvement with the project.

## 3. Results

### 3.1. Impacts on Suicidality

The CAMS research project commenced in April 2018 and has been embedded within services since the start of 2019. Preliminary data indicated a reduction in local suicide rates based on the judicial and clinical data sources (Figure 1) for residents of North East Lincolnshire and the subgroup of mental health patients, the latter being defined as individuals who had contact with mental health services in the 12 months prior to suicide [9]. These promising albeit mid-term results from the project required confirmation through coroner inquest verdict collation at a national level, to meet the established UK legal standard for death by suicide. The recent change to the standard of proof for the coroner’s court to reach a conclusion of suicide, from the criminal standard to the civil standard, might lead to higher figures from 2019 onwards [42]. The impact on suicide rates should, therefore, be regarded as tentative until the completion of the project.

At 24 months of the research project, 43% of service users presenting to crisis/mental health liaison services had had a “suicide risk triage” (total *n* = 2993), with 2% requiring CAMS intervention (*n* = 60). It is acknowledged that a vast literature demonstrates the effectiveness of CAMS for suicidal ideation [35,37], not just life-threatening behaviours, although this research focused on the latter.

The results highlighted that over half of the presentations to the crisis team were not related to suicide risk, which was unsurprising, given the nature of a 24/7 open access crisis service where any individual can present without a referral and in many cases, solely based on their own belief that they may require mental health support. It was our understanding that this was a unique and singular feature of this service provision within the NHS in England, given the crisis/mental health liaison teams have access to inpatient psychiatric beds and provide a crisis and home treatment service. This truly “open access” health and social care approach, unsurprisingly, resulted in only 0.86% of all such referrals to crisis/mental health liaison services presenting with acute suicidality (potential life-threatening behaviour), requiring fast-track access to inpatient services and the CAMS intervention.

### 3.2. Impacts on Service User Outcomes

Early quantitative data suggested a positive impact of a systems-level “suicide risk triage” model on several key outcomes of service utilisation. Table 1 depicts the means for the number of crisis/mental health liaison referrals, mental health inpatient admissions and attended appointments with mental health services, for service users 6 months pre and post their “suicide risk triage” where 6 months of follow-up data are available. Thus, individuals who had no history of crisis contact and presented for the first time to services during this period were excluded, as were those who had yet to reach the 6-months post-triage follow-up point. This included cases both with and without pre-existing mental health issues as due to the open-access nature of the services, a mental health diagnosis was not required for referral.

The difference between service utilisation 6 months pre and post “suicide risk triage” was assessed. Shapiro-Wilk tests highlighted that the assumption of parametricity was not met (*p* < 0.005); hence a non-parametric test was employed. A Wilcoxon Signed Ranks Test revealed statistically significant differences across service utilisation pre and post “suicide risk triage” (Table 1). There was a significant reduction in crisis/mental health liaison referrals (z = −20.711, *p* < 0.001), inpatient admissions (z = −7.462, *p* < 0.001) and inpatient length of stay in days (z = −5.300, *p* < 0.001), as well as significantly improved attendance at appointments (z = −3.893, *p* < 0.001).

In relation to completed suicides for this cohort, NHS policy and United Kingdom Coroner’s Court judicial procedures require a thorough root-cause analysis-based investigation of any death by suicide where the individual has had contact in the previous 12 months with an NHS service or equivalent organisation. Anecdotally and at the time of writing, there were 2 completed suicides since the commencement of the research project, although this number might be impacted significantly before its completion by the impact of the Covid-19 pandemic on suicidality and resultant increased presentations to mental health services.

In conjunction with the reduction in suicide during this period (Figure 1), these tentative results suggested that implementation of the “suicide risk triage” model improved clinicians’ objective decision-making, with services users being offered/signposted to more appropriate services to reduce their suicide risk, such that there was less need for ongoing or future crisis-level help-seeking behaviour or inpatient admission. Further, offering this individualised and “triage-informed” support for suicidality increased attendance at resulting mental health appointments, which would suggest that the model enhanced user engagement and recognition of the value of the planned clinical services they were offered following “suicide risk triage”.

### 3.3. Qualitative Data

The impact of service changes on service users and clinicians was assessed through qualitative interviews. These were conducted 18 months after the start of the project. Seven service users who had received the CAMS intervention were interviewed by a Research Associate. Questions were related primarily to their views about the CAMS intervention, how it impacted their clinical experience and any improvements that could be made for them and other service users presenting with suicidality.

A total of seven clinicians also completed interviews with an external clinician; four clinicians were CAMS trained, two were awaiting CAMS concordance supervision, and one was a suicide triage team supervisor (not CAMS trained). Clinicians were asked about their experience working with suicidal patients, their use of CAMS and suicide risk triage frameworks, any changes to their confidence or professional practice and any barriers to implementation.

The qualitative data collated from interviews would be discussed in a related paper, exploring the acceptability of CAMS for service users and changes to clinicians’ confidence since the implementation of the CAMS framework and associated training.

## 4. Discussion

Despite a clear downward trend in service utilisation for individuals presenting to crisis services with suicidality over the first 12 months of the project, caution is advised in interpreting these results before the completion of the project in 2022. A maintained 3-year trend for the lowered suicide rate would support the initial findings, and longitudinal follow-up data will be evaluated at the end of the project. However, clinical trial data is required for causal inferences to be made, and this is currently being discussed with other NHS providers. Nevertheless, mid-term results indicated that with sufficient training and strong leadership, implementation of a bespoke systems-level “suicide risk triage” model had the potential to impact on service user outcomes for both mental health and non-mental health cases. Several key areas of practice, including clinicians’ confidence, effective care planning and targeted treatment provision, could also be improved using this approach. The key strength of the “suicide risk triage” model was the ability to assess and intervene to determine appropriate support for suicidality presentations by focusing on variables pertinent to suicide risk, including non-mental health needs—the indirect drivers of suicidality. Further prospective follow-up data will determine whether there is a significant impact on suicide rates for non-mental health cases presenting to crisis services. Although the availability of open-access crisis support has been a highly promoted component of local mental health services, the challenge remains to improve help-seeking in certain cohorts who do not access this provision due to stigma, e.g., middle-aged men [43]. The data strongly suggested that providing such a service for mental health cohorts alone would not impact on deaths by suicide, where indirect drivers, such as negative life-events, lead to rapid suicidality, self-harm and acute distress in non-mental health populations.

Sustainment of the initiative was a key part of the project’s ongoing success at the mid-term. A plan for ongoing training of new clinicians and refresher training for those practising CAMS was implemented. Future plans include ensuring that departmental champions are CAMS-trained to improve their understanding of the CAMS intervention across all other services. Further measures of clinicians’ confidence will ascertain the extent to which suicide-specific training, utilising an evidence-based intervention, has benefits for the organisation and broader NHS, beyond solely that is observed in the crisis/mental health liaison teams.

Some lessons learned included the importance of the supervision structure around CAMS and triage for the successful implementation of the model. Support from the supervision hierarchy was integral to ensuring that any managerial pessimism towards a new model reverted to confidence as their staff teams were able to access, advise and support their colleagues. Implementation of any such initiatives dealing with risk issues needs to be supported at the executive level to ensure that key changes to clinical practice and adherence to the project are disseminated organisation-wide. Further, having project leads accessible within the crisis team ensured that support was on-hand when clinicians, who were new to CAMS, needed advice around challenging risk cases. Departmental champions were also a key link between the project leads and community teams so that any issues from within these teams could be discussed at the monthly meetings. Recent discussions at these meetings indicated that very few issues regarding the use of “suicide risk triage” were being raised by these teams, and this might be a result of the low numbers of suicides and a confident crisis/home treatment and mental health liaison service available to primary and secondary care teams without delay. Anecdotally, clinician retention and team morale were also improved. The anxiety clinicians experienced when assessing such high-risk cases reverted to confidence as they gained a full understanding of suicidality, which helped them successfully engage with the service user. Clinicians expressed feeling comfortable in seeking supervision, adhering to the hierarchical process, without fear of competency issues being highlighted and in both routine formal/informal clinical practice.

Further, the training structure was an important part of the way in which clinicians approached suicide risk assessment. Evaluation feedback from the initial ‘risk triage training’ suggested that experienced clinicians tended to use a positive risk-taking approach, whereas recently-qualified clinicians did not feel as confident with suicidality cases unless they were routinely confronted with such cases (such as those working in the crisis team). Clinicians from community teams commented on the benefit of having a supervision structure in place and an organisation-wide responsibility for dealing with challenging suicide risk cases.

In terms of the clinical use of CAMS, some wider benefits of the approach were observed by CAMS-trained clinicians. For instance, some CAMS-trained clinicians used questions from the SSF core assessment to frame their crisis assessments as a means of differentiating between different levels of clinical risk during their initial contacts with service users. Additionally, enthusiasm from clinicians outside of the crisis team who had not received the CAMS training was noteworthy. Some clinicians who had attended the ‘risk triage training’ approached the project team to learn more about CAMS as they felt they would benefit from having access to a suicide-specific framework. Several also asked if they could be CAMS trained.

## 5. Conclusions

Preliminary results in 2019 depicted a reduction in suicide numbers (prior to ratification), although caution is advisable, given that a 12-month trend would need to be continued into 2021/2022. The findings presented gave an indication that a systems-level “suicide risk triage” model could have a powerful impact on how NHS mental health trusts tackle the national problem of suicide prevention by targeting treatment based on the acuity of suicide risk and patient-defined drivers. The changes to service utilisation reflected not only the clinical cohort of those with mental health problems but also the general population within the catchment area—one of the most deprived in the UK. Additionally, clinicians’ confidence, positive risk-taking and effective treatment/care planning were all positives that were being observed.

There is a need for longer-term follow-up of the successes observed. A follow-up paper detailing the impact of CAMS on service user outcomes at follow-up, compared with previous “treatment-as-usual” using a historical matched control group, is in preparation. Analysis of whether CAMS is cost-effective in the long-term will be explored through further post hoc analyses and longitudinal study. Nevertheless, the positives outlined represent how a systems-level approach using CAMS assessment and intervention may be implemented within the NHS and can impact clinical processes for mitigating suicide risk. These mid-project findings also suggest that using CAMS can improve efficient provision in a difficult area of clinical practice. Future research will disseminate whether CAMS continues to reduce suicide, associated risks and the “core ingredients” for future NHS service utilisation within this high-risk population.

## Figures and Tables

**Figure 1 ijerph-17-04920-f001:**
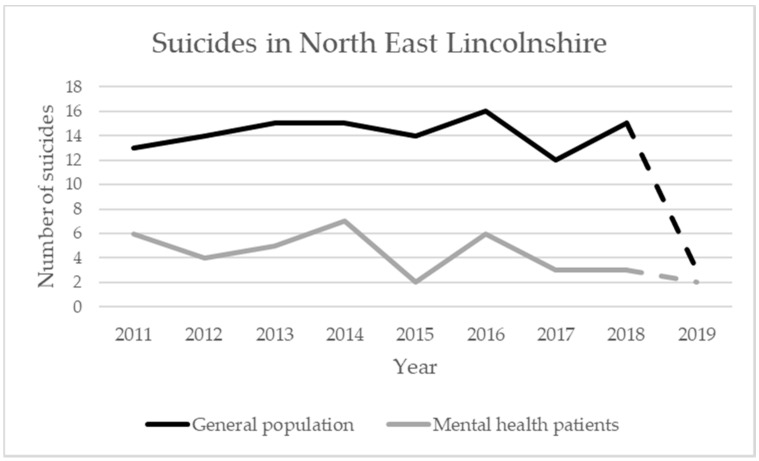
Suicides in North East Lincolnshire 2011–2019 (2018/19 data subject to ratification). The continuous line represents confirmed deaths by suicide, whereas dashed lines for years 2018/19 are tentative until legally confirmed via judgement at Coroner’s Court where conclusions of death by suicide are formally established in the United Kingdom.

**Table 1 ijerph-17-04920-t001:** An average number of crisis/mental health liaison referrals, inpatient admissions, inpatient length of stay and attended appointments (*n* = 782).

Outcome	6 Months Pre M (SD)	6 Months Post M (SD)	Test of Association
Crisis/mental health liaison referrals	1.43 (1.09)	0.35 (1.12)	z = −20.711, *p* < 0.001
Inpatient admissions	0.19 (0.50)	0.05 (0.24)	z = −7.462, *p* < 0.001
Inpatient length of stay (days)	3.50 (13.29)	1.31 (8.38)	z = −5.300, *p* < 0.001
Attended appointments	4.47 (7.89)	6.47 (12.83)	z = −3.893, *p* < 0.001

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
