# Peer review of "Embedding an Evidence-Based Model for Suicide Prevention in the National Health Service: A Service Improvement Initiative"

_ijerph, 2020, doi:10.3390/ijerph17144920_

Round 1
Reviewer 1 Report
Thank you for an opportunity to review the manuscript. This text is well written and provides valuable information and insight regarding clinical practice of suicide risk assessment and management. The content seems to be 'squeezed' into the journal template as it does not present a typical study, consequently the Method and Results sections are pretty narrative. Nonetheless, the narrative is engaging and worth reading.
Several suggestions for the authors:
Line 74 - please define 'drivers'
Line 76 - please, provide more information about the Collaborative Assessment and Management of Suicidality (CAMS); what does it encompass, how does it work?
Line 83 - please clarify "Given the issues described above"
Line 108 - please clarify meaning of 'base' in "The research was undertaken at the base for crisis, home treatment and acute services."
Lines 136-142 - I suggest providing more information on the 'hierarchy' earlier in the text (now section 2.4)
Line 149 - please provide more information re 'focus groups' and Line 160 on 'pre and post training surveys'
Figure 1. - please clarify the two types of lines - continued and dotted (since 2018) in the graph
Lines 270-272 and 315-322 - I suggest caution in concluding that the program had (immediate and direct) impact on suicide rates
Have the authors collected any qualitative data from patients on their experience of CAMS, triage etc. (see line 260 on interviews with clinicians)?
Author Response
Thank you for taking the time to review the manuscript and for the valuable feedback provided. Please see below a list of revisions to address/clarify the suggestions made:
Reviewer 1
- Additions made to define suicidal drivers (lines 96-97) and within description of the CAMS intervention (lines 101-114)
- Description of the CAMS intervention including the process and follow-up added (lines 101-114)
- The phrase “Given the issues described above” (line 126) relates to the futility of suicide risk prediction and related impact on clinician confidence. Extra information has been added to clarify (lines 124-128) and reference for issues around clinician confidence added (line 125)
- “Base” refers to hospital site, this has been amended in the text (lines 167-168)
- Information around the supervision hierarchy added earlier in the text – see two additional sections (lines 202-207 and 211-214)
- Explanation of focus groups added (lines 229-233), and information/reference added for the pre/post surveys (lines 243-245)
- Extra information added to figure caption to explain types of line (lines 317-319)
- Changes made to Discussion and Conclusion sections to clarify that the primary impact is on service utilisation, with longitudinal follow-up needed to assess changes to suicide rates (lines 383-388 and 444-449)
- Service user qualitative data has also been collected – this has been added in in the text (lines 368-373) and will be analysed in a follow-up paper
Reviewer 2 Report
Thank you for allowing me to read this interesting manuscript. It is very well written and pertains an important topic, which makes the content of your work very worthwhile and necessary for mental health practitioners. However, I do have some remarks, and I hope you can address them to strengthen the work further:
- The research you are presenting and the context you are describing are the UK context (and England in particular); perhaps it is wise to explain a bit more about the (mental) healthcare system in the UK for readers unfamiliar with it. The paper is very well written but seems at times to assume pre-existing knowledge; for example, what is an NHS Mental Health Trust and how do they relate to local healthcare providers?
- On page 2, you write "...and that the use of such scales may result in
48 unnecessarily restrictive treatment for those categorised as “high-risk”: what are these treatments, and why do these scales cause such treatment? Is it the case that people are incorrectly classified as high risk? - In line 49/50 you write "..An additional challenge for suicide prevention is appropriate provision for those with mental health and non-mental health related needs [16]. --> I assume you mean that the approach to help these individuals differs depending on whether they have (pre-existing) mental health issues or not. But you could make this a bit clearer by rephrasing this sentence or focusing solely on the non-mental health related needs.
- Sentence 68/69: "The structure forms an “aide memoire” of items characteristic of many suicide risk prediction tools that may provide a false reassurance of safe and effective working." --> please explain. What do you mean here with 'the structure', and why is an 'aide memoire' not helpful to them?
- I would like to have more information on what CAMS consistitutes: what are the elements, how long does it last etc.
- From line 83 onward, you seem to focus on the current research; it would help if you could use a subheading to indicate this. Furthermore, what is meant with 'The organisation' in line 83?
- Am I correct in assuming that the suicide risk triage you are discussing precedes the CAMS model, and as such is part of a comprehensive intervention to first assess suicide risk and second, treat/intervene accordingly? If so, how then is it possible to separate the effects of only the triage phase on suicidality, suicide rates, and service user outcomes (line 91)? I understand that the methodology will explain more, but it would help the reader to already know more about the interrelationships between the models you refer to in the last paragraphs of the introduction (line 83 onward).
- Method: Can you indicate how many individuals present at the open access crisis and home treatment service/team on a yearly (and/or monthly) basis?
- In general I feel I am lacking sufficient detail in the method section to understand the context in which this research was conducted. Especially the description of the site and the 'triage' intervention itself can do with a bit more explanation. For example: how big was the team (method, 1st paragraph)?
- From the explanation provided in 2.3, the CAMS seems to be part of the 'risk triage' - is this indeed the case? or are they part of a larger program?
- After reading the methodology, it still is not clear to me what the dependent variables are, i.e., what will be measured to determine whether the intervention was a success or not. Were there any pre-implementation measurements conducted on the core variables of interest? Were the same measures administered after the trial ended? To sum up, here as well, more detail is necessary to understand not only the context and the intervention, but also what the current research consisted of - this is largely lacking.
- Results: here, things become clearer in terms of what was being measured (and how). However there is still room for clarification. For example, section 3.2 details the impacts on service user outcomes. Here, the authors report on data 6 months pre-triage for n = 782 individuals. Am I correct to assume that these are then considered individuals with pre-existing mental health issues? Is there any data on how many of these individuals eventually committed suicide?
- The figure depicts changes in suicides - is it possible at all to conduct statistical tests on these data? I cannot imagine that there is a significant difference for the mental health cases (a drop from 2.5 or 2 to 2?). This then also puts into question the conclusion drawn in the first paragraph of the discussion ( "l has the potential to impact on suicide rates for both mental health and non-mental health cases."). I think main strength (and novelty) of the risk triage is its potential to 'catch' the non-mental health cases and to intervene in time, focusing on variables that perhaps are not only related to 'disorders'.
- After reading the discussion, the issue that remains for me is that I am confused as to how your 'risk triage' fits into the CAMS, whether the implementation of the CAMS is novel for the UK or whether it is the addition of the risk triage that is the novel element that you are evaluating here.
All in all, this is an interesting and promising approach to reduce suicides and to intervene at the earliest possible time. Clarification of some of the issues makes this paper even more relevant for both researchers and clinicians.
Author Response
Thank you for taking the time to review the manuscript and for the valuable feedback provided. Please see below a list of revisions to address/clarify the suggestions made:
Reviewer 2
- Details added regarding definition of mental health trusts and services provided (lines 41-47) as well as description of Crisis Resolution Home Treatment Teams (lines 47-50)
- Clarity added around treatments offered for those as “high-risk” and evidence from the literature of incorrect categorisation (lines 57-62)
- Sentence reworded to clarify the focus on non-mental health related needs (lines 63-65)
- The structure refers to the “checklist” style risk assessment tools used within NHS mental health services – clarified and added why “aide memoire” is not helpful (lines 84-91)
- Description of CAMS intervention added (lines 101-114)
- Subheading ‘Current research’ added (line 123); organisation changed to ‘mental health provider’ (line 127)
- It is correct that “suicide risk triage” precedes the CAMS intervention. Clarity added around this point added and how it fits with the CAMS intervention (lines 130-145)
- Number of referrals to crisis/home treatment added (line 168)
- Clarity around research site provided (lines 167-168), size of team (lines 173 and 175), assessment provided (lines 177-179) and description of “suicide risk triage” (lines 194-201)
- Clarified that the CAMS intervention is part of the triage model but represents a small number of cases presenting to services (lines 253-256)
- Outcome measures explicitly stated (lines 295-302). Contextual information relating to the team and triage model added in relation to previous points
- Clarity around what the n represents (lines 340-342) and numbers that completed suicide has been added (lines 352-358)
- Amendments made to the key findings from the project in the Discussion section (lines 383-388 and 393-400) and change made around impact on suicide rates to impact on service utilisation (line 390)
- Information around novel element of the research project i.e. CAMS intervention in the UK (lines 120-122)